# Improving COVID-19 Vaccine Uptake among Black Populations: A Systematic Review of Strategies

**DOI:** 10.3390/ijerph191911971

**Published:** 2022-09-22

**Authors:** Morolake Adeagbo, Mary Olukotun, Salwa Musa, Dominic Alaazi, Upton Allen, Andre M. N. Renzaho, Ato Sekyi-Otu, Bukola Salami

**Affiliations:** 1Faculty of Nursing, University of Alberta, Edmonton, AB T6G 2R3, Canada; 2Department of Pediatrics, University of Toronto, Toronto, ON M5G 1X8, Canada; 3Translational Health Research Institute, School of Medicine, Western Sydney University, Sydney, NSW 2560, Australia; 4Maternal, Child and Adolescent Health Program, Burnet Institute, Melbourne, VIC 3004, Australia; 5Department of Surgery, University of Toronto, Toronto, ON M5S 2J7, Canada

**Keywords:** COVID-19, Blacks, vaccine uptake, hesitancy, strategies, interventions

## Abstract

Given the growing body of evidence on COVID-19 vaccine hesitancy among Black populations, the aim of this systematic review was to identify the interventions and strategies used to improve COVID-19 vaccine confidence and uptake among Black populations globally. To identify relevant studies, we conducted a systematic review of the literature based on a systematic search of 10 electronic databases: MEDLINE, Embase, PsycINFO, CINAHL, Scopus, Cochrane Library, Web of Science, Sociological Abstracts, Dissertations and Theses Global, and SocINDEX. We screened a total of 1728 records and included 14 peer-reviewed interventional studies that were conducted to address COVID-19 vaccine hesitancy among Black populations. A critical appraisal of the included studies was performed using the Newcastle-Ottawa Quality Assessment Scale. The intervention strategies for increasing COVID-19 vaccine uptake were synthesized into three major categories: communication and information-based interventions, mandate-based interventions, and incentive-based interventions. Interventions that incorporated communication, community engagement, and culturally inclusive resources significantly improved vaccine uptake among Black populations, while incentive- and mandate-based interventions had less impact. Overall, this systematic review revealed that consideration of the sociocultural, historical, and political contexts of Black populations is important, but tailored interventions that integrate culture-affirming strategies are more likely to decrease COVID-19 vaccine hesitancy and increase uptake among Black populations.

## 1. Introduction

The global COVID-19 outbreak has significantly impacted human health. This disease, which was declared a pandemic on 11 March 2020, by the World Health Organization (WHO), not only increased population morbidity and mortality but also overburdened the global economy and public health systems worldwide. According to a May 2022 report, over 513,955,910 individuals have had COVID-19 with 6,249,700 related deaths worldwide [1]. With over 213 million confirmed cases, Europe was severely affected by COVID-19 [1]. Statistical reports in other countries also continue to indicate a high prevalence of the disease. As of May 2022, the United States (US) had recorded over 84 million cases, while Canada had reported 3.7 million confirmed cases that resulted in 39,000 deaths [1]. However, Africa recorded fewer cases of infection and extant mortality compared to other continents as 7.9 million cases resulted in 169,072 deaths [1].

The COVID-19 pandemic has contributed to the expansion of existing health inequalities within and between countries. Outside Africa, Black populations have been disproportionately affected; specifically, they are more likely than non-Black populations to contract SARS-CoV-2 and die from COVID-19-related complications [2,3]. A meta-analysis of the prevalence of COVID-19 among racialized groups worldwide reported higher rates of hospitalization and death among Blacks compared to White counterparts [4]. Similarly, the infection rate was three times higher and death rates six times higher in predominantly Black vs. predominantly White US counties [5]. Studies conducted in Canada report similar findings [6,7,8]. Specifically, communities comprising a higher population of Black residents experienced more COVID-19cases [7]; for instance, despite constituting 9% of Toronto’s population, reports indicate that Blacks accounted for 26% of all COVID-19 cases [9].

To mitigate COVID-19-related morbidity and mortality, several COVID-19 vaccines were developed, trialed, and rolled out. Several countries have reported variance in vaccine intention and uptake among ethno-racial groups. In April 2020, the US reported that an average of 42.5% of Black individuals were willing to be vaccinated compared to 51.5% of their White counterparts [10,11]. Similarly, approximately 66% of White Canadians had received at least one COVID-19 vaccine dose compared to only 45% of Black Canadians, as of June 2021 [12]. In Africa, existing data show the lowest rate of vaccination compared to any other region in the world as less than 25% of the entire population have either received at least one dose or are fully vaccinated [1]. Vaccination coverage in countries such as the US reveals marked disparities between Black and non-Black populations [13]. Despite a general increase in the proportion of Americans willing to receive the COVID-19 vaccine, only 57% of Black Americans had received at least one vaccine dose as of April 2022, compared to 63%, 65%, and 85% of White, Hispanic, and Asian Americans, respectively [13]. Although the gap in COVID-19 vaccine uptake between Black Americans and other ethno-racial groups has narrowed since the initial rollout of the vaccines, vaccination coverage among the Black population has consistently lagged and plateaued in recent months, while other groups continue to see substantial increases in total coverage [13].

Multiple factors influence vaccination intention and decision among Black populations. One such factor is the historical trauma related to unethical medical experimentation in the US and other countries such as Nigeria [14,15]. Contemporary encounters with medical racism and discrimination at the provider level have continued to erode trust between Black patients and healthcare providers [16,17]. In addition, the rapid development and approval of COVID-19 vaccines and their swift authorization for emergency use have heightened vaccine hesitancy and concerns over vaccine safety and efficacy in Black communities worldwide [14,18,19,20]. Inequities related to the convenience and cost of accessing COVID-19 vaccines and healthcare, in general, have contributed to vaccine hesitancy among Black populations [21]. Furthermore, Black people in Western countries are over-represented in essential service occupations, precarious employment, and neighbourhoods with crowded living conditions [22]. Once infected with SARS-CoV-2, Black people have higher mortality rates due to barriers to accessing healthcare and a greater burden of existing chronic diseases [23,24].

COVID-19 vaccine hesitancy remains a complex issue as it constitutes a barrier to vaccination goals, health, and general well-being [25]. Despite this widespread knowledge, vaccine hesitancy in Black populations is persistently high in the US, Canada, and United Kingdom, while trending lower in sub-Saharan Africa [12,26,27,28]. Numerous strategies and interventions have been implemented to reduce hesitancy and improve COVID-19 vaccine uptake in various countries and among specific ethno-racial groups, some of which, over time, have impacted the intent to vaccinate among Black populations [29,30,31]. However, the persistent and ongoing disparities suggest a need for more tailored and effective interventions to increase vaccine uptake among Black populations [31]. Knowledge is lacking with respect to the most effective interventions to address COVID-19 vaccine hesitancy among Black populations. Our systematic review sought to address this gap by identifying strategies utilized to support the uptake of COVID-19 vaccines among Black populations globally, with the goal to inform evidence-based strategies for improving COVID-19 vaccine confidence and uptake in this population. This paper utilizes the terms strategy and intervention interchangeably to highlight the different formal and informal approaches, methods, and programs implemented to improve COVID-19 vaccine hesitancy among Blacks.

With the prolonged and ongoing threat of COVID-19, this systematic review is timely and supportive of current public health efforts to reduce the spread of the disease. Improving COVID-19 vaccination rates among Black populations is of great interest to policymakers, healthcare providers, Black communities, and stakeholders, given the need for tailored interventions to combat vaccine hesitancy and address the multifaceted and historical factors underlying this growing phenomenon.

## 2. Materials and Methods

Our review addressed the following research question: “What intervention strategies have been used to address COVID-19 vaccine hesitancy among Black populations across the globe and what is the effectiveness of these strategies or interventions?”. The review describes the different interventions used to improve COVID-19 vaccine uptake among Black populations, the effectiveness and outcomes of these interventions, and future research directions that can help to address COVID-19 vaccine hesitancy and other health-related disparities.

### 2.1. Inclusion Criteria

Guided by the Preferred Reporting Items for Systematic Reviews and Meta-Analyses (PRISMA) statement [32,33], the articles included were those that: (a) addressed COVID-19 vaccine hesitancy and uptake among Black populations; (b) described an intervention or strategy for reducing COVID-19 vaccine hesitancy or described an intervention for improving COVID-19 vaccine confidence and uptake in Black communities; and (c) were based on a peer-reviewed empirical study.

### 2.2. Search Strategy

With the collaborative assistance of a university librarian, we searched through December 2021 for relevant peer-reviewed studies in 10 electronic databases: MEDLINE, Embase, PsycINFO, CINAHL, Scopus, Cochrane Library, Web of Science, Sociological Abstracts, Dissertations and Theses Global, and SocINDEX. The keywords used in the database search are presented in Table 1.

The search was not restricted by publication date, publication type, language, strategy, or intervention type. This broad search approach allowed for the capture of a wide range of interventions aimed at increasing COVID-19 vaccine uptake among Black populations. In addition to the database search, the reference lists of relevant articles were manually reviewed to broaden the scope and avoid publication and source selection bias.

The study records were retrieved and exported to Covidence, an online platform that supports the systematic review process, including article selection [34]. The records were individually screened by two reviewers against the set of inclusion criteria, first by title and abstract and subsequently by full text. Disagreements were resolved by discussion with two additional reviewers.

### 2.3. Data Extraction

A data extraction spreadsheet was created in Excel and pretested by two of the authors. Guided by the study objective and inclusion criteria, a standardized form was developed following the PICOT framework to extract the following elements from each of the included studies: (a) reference (author and year of publication); (b) country; (c) study design; (d) sample; (e) intervention used; (f) measurement scale; and (g) results (Table 2). Data extraction was completed by two members of the team and checked by two other reviewers.

### 2.4. Quality Assessment

The Newcastle-Ottawa Quality Assessment Scale [48] was adopted to assess the overall quality of each included study. This tool assesses the overall quality of case–control and cohort studies based on three important domains: (a) selection of the study groups, (b) comparability between the groups, and (c) confirmation of the exposure and outcome of the study group. Stars are awarded for each sub-standard if the criteria are met. Each study is allocated a total score that ranges from 0 to 9. Scores ranging from 7 to 9 indicate a good or higher level of quality, while studies with scores below 4 are considered to have a greater risk of bias.

### 2.5. Data Synthesis

Following the Preferred Reporting Items for Systematic Reviews and Meta-analyses (PRISMA) flow diagram structure, the search report and the tabulated data extracted for each included study were completed and are presented in this review. This review did not conduct a statistical grouping due to the variation in study designs, interventions and strategies used, sample populations, and results of included studies. Rather, the key findings from each study were narratively synthesized in accordance with the objectives of this systematic review. The narrative synthesis provides a detailed and robust connection between the studies.

## 3. Results

A total of 1728 records were identified and retrieved from multiple databases, with 1019 duplicates removed prior to screening. The remaining records (*n* = 709) were individually screened by two reviewers against the set of inclusion criteria, first by title and abstract and subsequently by full text. The title and abstract screening resulted in the exclusion of 532 studies. The full-text screening of the remaining records (*n* = 176) led to the exclusion of 164 additional studies. Twelve (12) studies met all of the inclusion criteria. Two additional studies were included after hand-searching the reference lists of the 12 included studies (Figure 1). All 14 included articles were peer-reviewed empirical studies.

Of the 14 included studies, 11 were quantitative but only five stipulated the research design used: cross-sectional surveys (*n =* 1), randomized survey experiments (*n =* 3), and quasi-experimental designs (*n =* 1). One study utilized a qualitative design, and another was based on a mixed-methods design. One study did not specify the design used. Most of the included studies were conducted in the US (*n* = 10), with the others conducted in South Africa (*n =* 2), Nigeria (*n =* 1), and Ethiopia (*n =* 1). These studies were published between March 2020 and December 2021.

### 3.1. Quality Appraisal

Overall, the quality of the included studies was good (Table 2). The quality of the three controlled intervention studies, quasi-experimental study, and cross-sectional study were rated as good (i.e., a score of 7 and above), while the remaining quantitative studies were rated fair due to their poor methodological rigor in the areas of sample representativeness, limited information on research designs, and statistical analysis. The qualitative study was rated as low quality. A detailed risk of bias assessment for each study is presented in Table 3.

The key intervention strategies utilized in each included study were narratively synthesized and categorized into three major groups: (i) communication and information-based interventions; (ii) mandate-based interventions; and (iii) incentive-based interventions. These themes are presented below.

### 3.2. Communication- and Information-Based Interventions

Interventions related to the utilization of communication and information to address COVID-19 vaccine hesitancy among Black populations were reported in 11 studies. Two studies assessed the impact of guidance and counselling delivered with standardized COVID-19 vaccination discussion tools and visual illustration communication tools on the intention to take COVID-19 vaccines [40,44]. Approaches such as clinical education on facts and information on COVID-19 vaccines by experts, peer influencers, religious and faith leaders, stakeholders, and direct engagement with community members were reported in seven studies [35,37,38,43,44,45,47]. Most of these strategies resulted in a substantial increase in participants’ COVID-19 vaccine intention and uptake. However, specific information such as vaccine safety and vaccine profiles had limited impact on participants’ vaccine uptake [45]. Moreover, four studies reported information dissemination through online platforms, webinars delivered by a culturally inclusive and representative group of healthcare professionals, and E-health educational media [35,38,41,46]. In the US, for example, members of the Diversity, Equity, and Inclusion (DEI) committee developed a social media educational campaign (“Vaccine Acceptance” page) that included resources about the COVID-19 vaccine in Spanish and English [38]. Collectively, the outcomes of the intervention indicate significantly reduced COVID-19 vaccine hesitancy among Blacks across the chosen sites. Evaluating how popular representations and communication of race/ethnic disparities regarding COVID-19 cases and deaths influence vaccine hesitancy, [39] reported that information that acknowledged historical racism had no significant impact on vaccine uptake among Black populations.

### 3.3. Mandate-Based Interventions

This review identified only one study [36] that reported a mandate-based intervention. Using a multicomponent survey and experiments, the intervention examined if establishing vaccine requirements could strengthen or weaken COVID-19 vaccine intention and uptake in the US. Randomly sampled and rotated participants were asked questions through different conditions (controlled and freedom). The outcomes of the intervention indicated an increased likelihood of vaccination among Black participants with the requirement condition; specifically, 80% of Black respondents reported the intent to vaccinate within the required condition vs. only 56% within the control condition.

### 3.4. Incentive-Based Interventions

Two studies reported incentive-based interventions. In [28], a randomized survey experiment design was implemented to examine the influence of financial incentives on COVID-19 vaccine uptake among Black populations. The study reported that large financial incentives were counterproductive in Black populations, as there was no increase in COVID-19 vaccine uptake. In [42], the economic and clinical outcomes of various approaches to COVID-19 vaccination programs in South Africa were estimated. The outcome of the study showed the different economic and clinical benefits of various vaccination strategies in different situations.

## 4. Discussion

The findings of this systematic review suggest multi-component interventions that integrate increased communication, culturally inclusive informational materials, community outreach, and greater accessibility are the most consistently effective. Mandate- and incentive-based interventions were less popular within our identified studies and also less consistently effective. This underscores the weight of informed choice, trust, and accessibility in vaccine uptake [49,50]. Specifically, the results of this systematic review indicate the availability of vaccines alone may not be sufficient to increase uptake in Black populations. This is evident in one study [40] in which no significant increase occurred in the number of pregnant women who chose to vaccinate based on availability at the primary care clinic. Considering that mistrust and misinformation are significant drivers of vaccine hesitancy in Black populations, vaccine availability must be integrated with measures to increase trust and provide clear information [51]. While [40] offered informational counseling to participants, the use of a standardized discussion tool may have limited their ability to tailor these interactions to the specific information needs of each patient, which has been documented to improve the reception of information in racial and ethnic minority populations [52].

In further considering the impact of vaccine availability, [41,42] show that availability and consistent engagement during vaccine campaigns may increase vaccination within members of an organization (e.g., in a healthcare setting), but the pace of vaccination should also be considered to maximize vaccination coverage while minimizing mortality and healthcare costs. In relation to this, community-oriented communication approaches are suggested to address concerns and potential misinformation related to the newness and effectiveness of the vaccine, side effects of vaccines and boosters, and changes in the public health information provided [52,53]. Communication and information pathways require consistent, transparent, tailored, and high-quality information that is clear and easy to understand [51]. For instance, [47] utilized translated SMS messages in addition to communications and workshops led by community leaders to align with these suggestions, contributing to increased vaccine acceptance in the community.

Consistent across several studies in our review was the emphasis on the need for engagement with trusted individuals within the community, such as faith leaders [35] or opinion leaders [47], to spearhead knowledge dissemination efforts. Recent reports accentuate the importance of building trust through community messengers, offering choice, providing social support, focusing on diversity in messaging, addressing misinformation, and providing tailored communication [26,49,52]. For instance, Black Americans are two times more likely to trust and listen to Black community leaders and messengers than messengers from White community [51]. In addition to engagement with community leaders, developing culturally inclusive materials and making vaccine information more accessible are also paramount [50]. Ref. [38] opined that information that is provided through culturally relevant messaging, using appropriate language for the community, and is sensitive to tone may generate more interest in COVID-19 vaccines. Similarly, related studies affirm that cultural discordance and language barriers between patients and healthcare providers impact the quality of patient–provider interactions, information shared, and the care received [54].

Communication of information from culturally representative healthcare professionals can have positive outcomes in terms of vaccine acceptance within Black populations [35,37,38]. Interactions with health professionals who are attentive to concerns and acknowledge the lived experiences of Black individuals highlight the importance of and desire for trusted information to guide decision-making or support vaccination intentions [43]. By acknowledging how lived experiences affect health and health decisions, the communication of health information is more attentive to the needs of communities and to how best to approach information sharing [44,46]. Though several resources aim to increase COVID-19 vaccine uptake in racial and ethnic minority populations, the needs of the communities these categories encompass are not identical. Targeted approaches to the provision of information may be more welcome by Black populations [39].

Consistent with a targeted and need-based approach, the importance of racial and social identity should not be underestimated. The sometimes complex dynamic between personal choice and duty or obligation is reflected in a few of the studies identified [28,36]. For example, past survey findings indicate Black Americans with a stronger sense of belonging within the Black community are more likely to report a willingness to vaccinate, as their sense of social responsibility to others was associated with an increased likelihood of vaccination [51]. A sense of responsibility, whether to others or related to the attainment of an objective, may be a noteworthy driver for vaccination among some Black individuals. Due to the COVID-19 vaccine-related hesitancy among Black populations, a requirement to vaccinate for work or travel may stimulate some individuals to vaccinate, whereas less intent is evident when given a choice [36]. However, the use of incentives to increase vaccination in Black populations should be employed with care, as incentives that are perceived as being too large may be detrimental to efforts to increase vaccine acceptance [28].

Overall, the conclusions made in our included studies were appropriate based on the designs and available data. Studies employing survey, questionnaire, or experimental approaches such as [28,36,39,44,45,46] which included controls and comparisons allowed the researchers to directly associate their variable of interest with the various outcomes. We see from these studies that incentives are effective but have their limits; mandates increase rates of intent to vaccinate; and communication and information-based approaches are more effective when they are culturally inclusive. These findings are reflected in other non-experimental studies as well, where targeted interventions were associated with increasing enrollment or vaccination counts [35,37,38,41,43]. Two studies where the conclusions may not be transferrable were the case of Hirshberg and colleagues’ simulation [40] and Yemer and colleagues’ [47] information campaign. A simulation based on an ideal scenario may not account for naturally occurring variances in human behavior. Additionally, while Yemer and colleagues [47] concluded that the use of various communication strategies contributed to greater vaccine advocacy and increased vaccine acceptance, they provide no true evaluation methods to support this finding. In studies with no control, it is also possible that other factors contributed to the increasing counts in vaccination intent or completion, such as simply having time to consider the vaccination or weigh options. Further study on this topic is thus required to support the development of standardized interventions.

In conducting this systematic review, some of the important strengths and weaknesses of the included studies and overall limitations cannot be overlooked. Though the studies were overwhelmingly quantitative, a strength of these studies was the moderate variety in the designs employed by the researchers, which collaboratively provide a broader perspective to this topic. The use of experimental approaches in [28,36,39,44,45] allowed for measurement of the effects of specific variables on intent to vaccinate; the use of surveys in [28,36,39,45,46] facilitated data collection from a large sample, ranging from 188 to 1353 participants among our included studies; and by employing a community-based participatory approach, Andrasik and colleagues [37] were able to meaningfully involve the community in the intervention, which directly supported their objectives of increased community engagement.

There were areas of weakness also noted across these studies. Our review revealed that there is a scarcity of research on this topic, particularly outside of the United States. Additionally, the heavily quantitative nature of the studies also highlights the need for greater understanding of the perspectives and lived experiences of Black communities regarding how and why certain interventions may be more or less effective. While most studies provided a detailed account of the interventions and evaluation processes, a few provided little information as to how their strategies were employed [38,39,41,42,47]. Finally, half of our included studies did not clearly identify a specific approach, posing a challenge in assessing for methodological congruence in the undertaking of the study.

## 5. Conclusions

This systematic review makes an important contribution to research by converging a diverse body of evidence that will influence future studies on COVID-19 and general vaccine uptake among Black populations. This systematic review of the extant literature on interventions to increase COVID-19 vaccine uptake in Black populations revealed a paucity of research on this topic, particularly outside of the United States. Specifically, an evident gap in relation to published and relevant interventions in many other countries, such as Canada, indicates that future research should inform knowledge by building on this review. The heavily quantitative nature of the studies also highlights the need for a better and in-depth understanding of the perspectives of Black communities regarding how and why certain interventions may be effective. The results of our review underscore the importance of interventions that attend to various social needs, are attuned to cultural values, and consider the socio-historical and political contexts of the Black community. Overall, the need to understand and re-evaluate the contemporary relationships of Black communities with public health and medical systems appears relevant with respect to addressing COVID-19 vaccines and vaccine-related hesitancy.

Though the intent to vaccinate has increased in Black American populations over time, the ongoing disparities in Western countries suggest the need for more strategic interventions to increase vaccine uptake in Black populations [31]. Our findings demonstrate the need for collaborative approaches that engage communities in identifying their priorities. Considering the importance of trust in the intent to vaccinate, community leaders should be involved at every stage of COVID-19 vaccination programs, from conceptualization to evaluation. Furthermore, in the process of developing interventions, public health officials need to consider issues related to confidence and access by asking guiding questions such as: Do people in the community want to be vaccinated? Can people in the community easily get vaccinated? How informed are people on vaccine-related issues to make informed decisions? Asking these questions supports decision-making about the type of interventions that are needed in a community [50]. These questions also bring attention to the various sociocultural factors that contribute to the rates of COVID-19 vaccination in Black communities.

## Figures and Tables

**Figure 1 ijerph-19-11971-f001:**
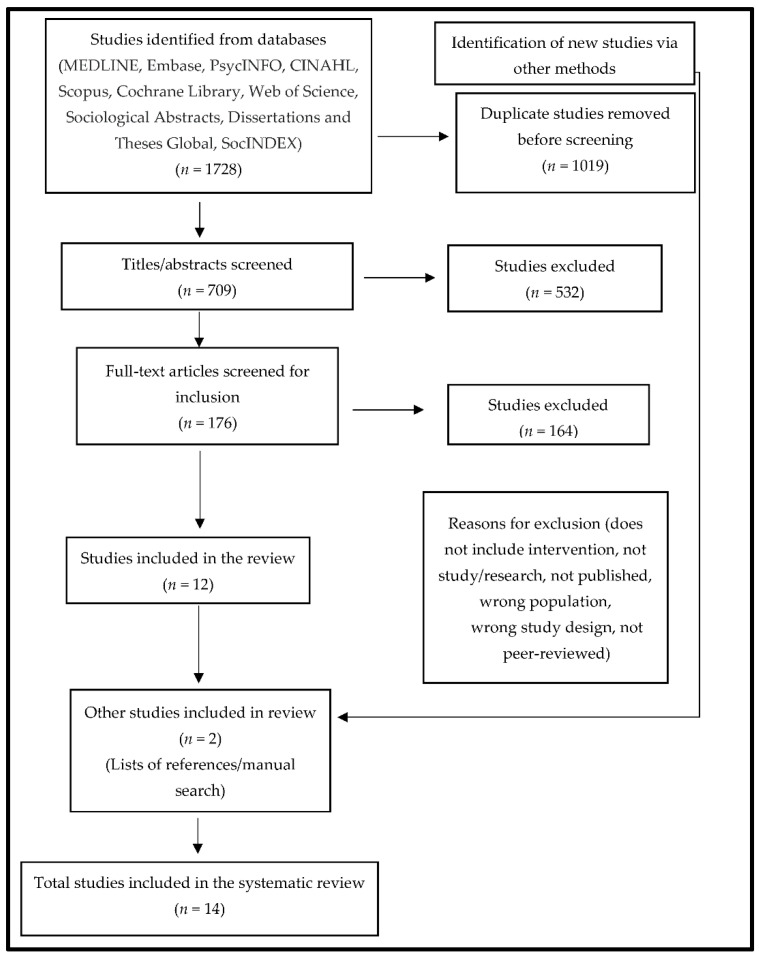
PRISMA flow diagram of the search strategy and included studies.

**Table 1 ijerph-19-11971-t001:** Key Search Terms.

Database	Search Terms
MEDLINE Embase PsycINFO CINAHL Scopus Cochrane Library Web of Science Sociological Abstracts Dissertations and Theses Global SocINDEX	African Continental Ancestry (black* or African* or Caribbean or afro* or “person of color” or “people of color” or colored or “dark-skin*” or BIPOC or ((racial or ethnic) minority* (Algeria* or Angola* or Benin* or Botswana* or “Burkina Faso” or Burundi or Cameroon or “Cape Verde” or “Central African Republic” or chad or Comoros or Congo* or “cote d’ivoire” or “Ivory Coast” or Djibouti or Egypt* or Guinea* or Eritrea or Ethiopia* or Gabon or Gambia* or Ghana* or Kenya* or Lesotho or Liberia* or Libya* or Madagascar or Malawi* or Mali* or Mauritania* or Mauritius or Morocco or Mozambique or Namibia* or Niger or Nigeria* or Rwanda* or “sao tome and Principe” or Senegal* or Seychelles or “Sierra leone” or Somalia* or “South Africa” or Sudan or Swaziland or Tanzania* or Togo or Tunisia* or Uganda* or Zambia* or Zimbabwe*) (Cuba* or “Dominican Republic” or Haiti* or Hispaniola* or “Puerto Rico” or “Puerto Rican*” or Jamaica* or Barbados or Dominica or Grenad* or “Saint Lucia” or Trinidad* or Bahama* or “Virgin Islands” or Anguilla or “Saint Kitts” or Antigua or “Turks and Caicos” or “West Indies” or “Saint Vincent”) Coronavirus or Coronavirus Infections or (coronavirus* or corona virus* or OC43 or NL63 or 229E or HKU1 or HCoV* or covid* or Sars-coronavirus* or Severe Acute Respiratory Syndrome Coronavirus*) and or Middle East respiratory syndrome or camel* or influenza virus or avian influenza or H1N1 or H5N1 or H5N6 or IBV or murine corona*) or Covid-19 or COVID-19 Vaccines and (confidence* or trust* or faith or accept* or uptake or hesitant* or attitude* or distrust* or mistrust* or reject* or refuse*) (vaccine* or immunize* or immunise*) or Vaccination Refusal

**Table 2 ijerph-19-11971-t002:** Characteristics of included studies.

Reference	Country	Study Design	Sample/Settings	Intervention	Measurement Scale/Instrument	Results	Quality Assessment
Abdul-Mutakabbir et al. (2021) [35]	United States	Not stated	General public—Mass vaccination clinic (*n* = 24,868); Mobile vaccination clinic (*n* = 1542); San Bernardino County (*n* = 2,180,085)	A three-tiered approach intervention: engagement with faith leaders in the academic community; a culturally inclusive and representative group of healthcare professionals from Loma Linda University to deliver the COVID-19 informational webinars; the completion of remote vaccination clinics.	Vaccination counts	Over the course of the intervention (February 1–30 April 2021), 24,808 persons received their vaccination at the routine mass vaccination clinic. 3.7% of these individuals were Black. At the remote clinics, 1542 individuals received a dose of the vaccine, with 44% of these individuals identifying as Black.	7
Albarracin et al. (2021) [36]	United States	Multicomponent study: 1 survey & 3 experiments (3-cell, within-subjects experimental design; 3 × 2 within-subjects factorial design; between-subjects design)	Total sample, randomly sampled US population including White, Black, & Hispanic American respondents (*n* = 1621); Survey (*n* = 299); Experiment 1 (*n* = 359); Experiment 2 (*n* = 357); Experiment 3 (*n* = 606). Randomly sampled US population	*Survey:* participants were asked two questions; *Experiment 1*: participants were randomly rotated through 3 different conditions; *Experiment 2*: participants were randomly rotated through 3 different conditions; *Experiment 3*: participants were randomized to either condition 1 or condition 3.	Yes/No scale, Likert scales	*Survey*: for Black participants, there was an increased likelihood of vaccination with the requirement questions than with the control question (*p* = 0.001); 80% of Black respondents reported the intent to vaccinate within the required condition; 56% of Black respondents reported the intent to vaccinate within the control condition. *Experiments 2, 3, 4*: data not disaggregated	9
Andrasik et al. (2021) [37]	United States	Community-based participatory research	Scientists and community leaders from four BIPOC communities (*n* = 40–60). Clinical trial sample size: variable throughout the course of study.	Meaningful involvement of the community; Stakeholder engagement and trust-building; Faith initiative: faith-based advisory; Communications & community influence	Pre-screening survey, enrollment counts	Across all 4 clinical trials, 47% of the enrolled participants were BIPOC, 15% of whom identified as Black or African American. The enrollment rates of various BIPOC communities aligned with their proportions within the broader US population. BIPOC enrollment was slow initially, but steadily grew and eventually overtook White enrollment. COVID-19 Prevention Network (CoVPN) sites that actively engaged with BIPOC communities had more success in recruiting BIPOC participants compared to non-CoVPN sites.	7
Feifer et al. (2021) [38]	United States	Not stated	Black and Hispanic healthcare workers/employees (*n* = 27,000)	Facts and informational sessions provided by experts; social media educational campaign; resources on the COVID-19 vaccine in Spanish and English; provision of culturally sensitive information through small group discussions and one-on-one conversations	Not stated	At the first time point, rates of vaccination were lowest among Black employees (45.5%) and highest among Asian employees (74.5%). The greatest increase in vaccination rates were among Alaskan Native employees (8.2%), Hispanic employees (6.1%), and Black employees (5.4%). There was a statistically significant increase in the likelihood of Black employees receiving the COVID-19 vaccination compared to White employees (*p* = 0.004).	6
Fox A. & Choi, Y. (2021) [39]	United States	Survey experiment	New York residents (*n* = 1353 (443 non-Hispanic Blacks, 429 non-Hispanic Whites, 481 Hispanics).	A messaging campaign that acknowledged historical racism through a newspaper prime that prioritizes minorities	Questionnaire	Participants’ intention to get vaccinated against COVID-19 was not influenced by the messaging campaign.	5
Hirshberg et al. (2021) [40]	United States	Not stated	High-risk obstetrical patients (*n* = 87)	Counseling on COVID-19 vaccination for pregnant and breastfeeding patients; onsite vaccine availability	Standardized COVID-19 vaccination discussion tool, state database	Patients seen before onsite vaccination was available: 1/32 (3%) patients received the vaccination offsite post-counselling. Patients who were seen after onsite vaccination: 2/55 (3%) received the vaccination onsite; 4/55 (7%) received the vaccination offsite. The availability of onsite vaccination was not significantly associated with an increase in vaccination (*p* = 0.22). Among the 55 patients who received counseling during the availability of onsite vaccination, 25 were counseled during the pilot program but did not receive a vaccination onsite or offsite.	7
Reddy et al. (2021) [41]	South Africa	Not stated	Healthcare workers (*n* = 7400)	Creation of staff database for appointment management; Vaccinators completed an online training program for healthcare workers; Vaccines were distributed by pharmacy teams and occupational health nurses. Reconciliation of doses was undertaken after every 24 doses dispensed. Staff arriving to be vaccinated had to display identification of booking confirmation, screened for symptoms, provided consent, then were observed for 15 min in a separate area after vaccination.	Dose counts	There was an upward trend in the number of vaccine doses administered in the week after the roll-out was initiated. Number of vaccines administered each subsequent day of the vaccination campaign (from 17 February 2021 to 26 February 2021, excluding 20 &21 February) = 32, 240, 460, 788, 823, 1018, 1138, 1160.	5
Reddy et al. (2021) [42]	South Africa	Not stated	South Africa’s population (specific sample size not stated)	COVID-19 vaccination program simulations over 360 days under various scenarios	Incremental cost-effectiveness ratio (ICER) = difference in healthcare costs divided by difference in years-of-life saved (YLS)	There were varying economic and clinical benefits to various vaccination strategies in the two scenarios—lack of vaccination program led to greatest amount of infections, deaths, costs; a 40% vaccination rate led to the lowest healthcare costs on both scenarios and decrease in deaths in both scenarios; a 67% vaccination rate also decreased deaths, however healthcare costs increased; an 80% vaccine supply and 80% vaccine acceptance decreased deaths and increased healthcare costs in both scenarios. Highest pace of vaccination at 300,000 daily led to optimal clinical outcomes and lowest costs when compared with lower paces in both scenarios. In the scenarios where the effective reproductive number = 1.4, 67% vaccine supply was less efficient (in terms of incremental cost-effectiveness ratio: the difference between healthcare costs divided by the difference between years-of-life saved, compared with other approaches to supply and pace) than the 80% vaccine supply. In the scenarios a two-wave epidemic, 67% vaccine supply was more efficient than the 80% vaccine supply; 20% vaccine supply was less efficient that higher levels of vaccine supply, yet decreased deaths by 72–76% and decreased healthcare costs by 15–32% when compared with no vaccination.	4
Robertson et al. (2021) [28]	United States	Randomized survey experiment (between-subjects design)	American adults (*n* = 1000)	Participants were assigned to four conditions for financial incentives ($1000, $1500, $2000, or no incentive).	Online survey form	Large financial incentives were found to be counterproductive in Black populations. 53% of Black Americans in the group receiving no incentives reported that they would most likely take the vaccine. The $1500 incentive resulted in an increase to 68% in the proportion of Black participants who reported that they would most likely take the vaccine. At the $2000 incentive, only 39% of Black respondents reported willingness to take the vaccine (a value 13.6% below the acceptance rate in the control group receiving no financial incentive).	7
Serper et al. (2021) [43]	United States	Not stated	Solid organ transplant recipients (*n* = 103)	Outreach & motivational interviewing by transplant center staff. Follow-up calls were scheduled to evaluate if participants proceeded to receive the vaccine.	Vaccination counts	People who identified as Black were more likely to schedule a vaccination appointment during the call with transplant center staff or to schedule a vaccine appointment on their own time compared to all other races (odd ratio = 1.6, 95%CI 1.02–2. 6; *p* = 0.042) after they received information from staff.	7
Ugwuoke et al. (2021) [44]	Nigeria	Quasi-experimental design	Victims of conflict in Nigeria (*n* = 470)	Artistic illustration communication intervention delivered concurrently with counseling on the acceptance of COVID-19 vaccination after it was available. This included experts from various disciplines: fine and applied arts, guidance and counseling, mass communication. It took place over the course of 10 days, with each session lasting 1 h. The control group did not receive this intervention. There was collaboration between the research team and the officials at the camp to ensure that the two groups did not interact over the course of the study.	Questionnaire	Participants who engaged with visual communication and counseling had greater average reports of self-efficacy than those who did not on the post-test questionnaire (3.3 vs. 1.4); they also had greater average reports of task efficacy on the post-test questionnaire (3.6 vs. 1.4). There were greater positive reports in terms of intent to vaccine in those who had exposure to visual arts on COVID-19 vaccination on the post-test questionnaire (3.8 vs. 1.3). Participants who had exposure to visual messages about COVID-19 vaccination reported higher levels of intention to vaccinate compared to the control group (self-efficacy 1.4 vs. 3.3, task efficacy 1.6 vs. 3.6, vaccination intention 1.1 vs. 3.8).	8
Wagner et al. (2021) [45]	United States	Randomized survey experiment	Detroit residents (*n* = 1117; 76.5% Black)	Hypothetical vaccine profile experiment	Questionnaire	More participants (77.1%) presented with a higher (95%) hypothetical effective vaccine had more acceptance rate compared to participants presented with vaccine that has lower (50%) effectiveness. Vaccine safety and some vaccine profiles had no significant impact on participants’ vaccine acceptance.	9
Williams-Gunpot (2021) [46]	United States	Cross-sectional survey	African American adults (*n* = 188)	E-health educational intervention that consisted of a pre and post knowledge test. The educational intervention (Our COVID-19 Knowledge Test) is a true-false tool programmed to have all true answers. The test consisted of 44 questions.	‘Our COVID-19 Knowledge’ Test (OCKT-44); COVID-19 Knowledge Scale; COVID-19 Prevention Self-Efficacy Scale; Intention to Vaccinate for COVID-19 (IVC-1) tool; Diffusion of Innovation of Our COVID-19 Knowledge Test (DOI-OCKT-1) Tool; COVID-19 Knowledge and Self-Efficacy for Risk Reduction Behaviors	Knowledge of COVID-19 was increased in the post-test (mean score 4.57 vs. 4.85, *p* = 0.000); knowledge of COVID-19 self-efficacy related to risk reduction (including vaccination) was also higher in the post-test (mean score 5.17 vs. 5.33, *p* = 0.000). 127 participants reported being vaccinated or having the intent to vaccinate. For the open-ended question, participants had feelings of hope about vaccination, noting that while there is still a chance of contracting COVID-19 with the vaccine, only a small percentage of vaccinated individuals get COVID-19.	8
					(C-K-SE-FRRB-4) tool		
Yemer et al. (2021) [47]	Ethiopia	Not stated	Leaders disseminating messages; sample size not stated	Various communication strategies and messages were analyzed. Messages such as health information about COVID-19 vaccination, vaccination campaigns, advocacy, and vaccine hesitancy. SMS messages were sent, religious leaders were involved to increase the spread of positive COVID-19 vaccine messages, workshops were developed to increase awareness of COVID-19 vaccines.	Not stated	Messages (TV, radio, newspaper, social media) about COVID-19 vaccine hesitancy reported on side effects (4/4), accuracy/low quality (2/4), safety testing (3/4), doubts about vaccines (4/4), and others (1/4). The use of various communication strategies contributed to greater vaccine advocacy and vaccine campaigns contributed to increased vaccine acceptance.	3

**Table 3 ijerph-19-11971-t003:** Quality Assessment of Included Studies using the Newcastle–Ottawa Scale (each study is allotted the stars (*) for each category (Selection, Comparability and Outcome/Exposure).

Study	Selection	Comparability	Outcome/Exposure	Total
[35]	****	*	**	7
[36]	*****	*	***	9
[37]	****	*	**	7
[38]	**	*	***	6
[39]	***	*	*	5
[40]	***	*	***	7
[41]	**	*	**	5
[42]	*	*	**	4
[28]	*****	*	*	7
[43]	***	*	***	7
[44]	*****	*	**	8
[45]	*****	*	***	9
[46]	*****	*	**	8
[47]		*	**	3

## Data Availability

Not applicable.

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
