# Peer review of "Improving COVID-19 Vaccine Uptake among Black Populations: A Systematic Review of Strategies"

_ijerph, 2022, doi:10.3390/ijerph191911971_

Round 1
Reviewer 1 Report
The study is about a meta-analysis investigating the effectiveness of interventional strategies which have been applied to improve COVID-19 vaccination in Black populations. The strategy to screen for included studies is clearly described and easy to follow. The result demonstrates how important effective communication and information dissemination are in terms of improving vaccination uptake in Black populations.
(1) It is preferred to see more emphasis on why this study is important in both "abstract" and "discussion" sections;
(2) I am wondering if any of those included interventional studies talked about the average education level of the population (participants).
Author Response
Thank you for your feedback on our manuscript. We appreciate your comments, time and effort in improving the manuscript for publication. Please see below our point-by-point response to each of your comments:
The study is about a meta-analysis investigating the effectiveness of interventional strategies which have been applied to improve COVID-19 vaccination in Black populations. The strategy to screen for included studies is clearly described and easy to follow. The result demonstrates how important effective communication and information dissemination are in terms of improving vaccination uptake in Black populations.
(1) It is preferred to see more emphasis on why this study is important in both "abstract" and "discussion" sections;
Response: Thank you for this suggestion. Due to the required number of words in the abstract section, we are unable to add more details, but we have extensively addressed the importance of the study in the discussion section. Please see lines 285-392
(2) I am wondering if any of those included interventional studies talked about the average education level of the population (participants).
Response: Some of the included studies may have referred to participants' level of education, but since this does not directly or necessarily impact the objectives of this review, limited attention was given to this factor.
Reviewer 2 Report
General Overview
The study presented a very significant issue in addressing COVID-19 vaccine hesitancy. Interventions are very fundamental, and this paper presents a systematic review of findings from previous interventions among Black populations.
Introduction
There are a lot of issues I observed in the introduction section of this paper. These observations also call to question the suitability of the title of the study. Firstly, I do not see understand the scope of this study given the fact that presented statistics are largely on the US and European countries. For paper that focused on Black populations, I was looking for statistics on COVID-19 prevalence, mortality and vaccination in African countries. It then seems to me that the study would be best focused on the Black populations in the western countries. Specifically, the arguments are not in complete harmony with reality on ground. African continent seems to be least affected by COVID-19. Therefore, the authors need to decide whether this study should focus on the black populations globally or be narrowed to those in western countries.
Representativeness of the Data
The issue of representativeness of the data is also fundamental. The study used 14 articles and only four were carried out in the core Black countries. This raises some concerns if the results can be generalized to the Black populations globally. It would have made more meaning if the authors had just used those 10 studies from the USA and revise the topic to focus on Black populations in the USA. It is also important to note that the data were collected from different population groups with only few conducted among the large populations. One then wonders if the larger Black populations behave in the same way as healthcare workers or scientists.
Results
In line 255 and 256, the authors reported significant increase in participants’ COVID-19 vaccine intention and uptake. Was there any statistical test carried out? How do we gauge the increase as being significant without a statistical test?
Will it not also be appropriate if the results are classified according to the nature of the respondents?
Conclusion
Present the limitations of the study.
Author Response
Thank you for your feedback on our manuscript. We appreciate your comments, time and effort in improving the manuscript for publication. Please see below our point-by-point response to each of your comments:
General Overview
The study presented a very significant issue in addressing COVID-19 vaccine hesitancy. Interventions are very fundamental, and this paper presents a systematic review of findings from previous interventions among Black populations.
Introduction
There are a lot of issues I observed in the introduction section of this paper. These observations also call to question the suitability of the title of the study. Firstly, I do not see understand the scope of this study given the fact that presented statistics are largely on the US and European countries. For paper that focused on Black populations, I was looking for statistics on COVID-19 prevalence, mortality and vaccination in African countries. It then seems to me that the study would be best focused on the Black populations in the western countries. Specifically, the arguments are not in complete harmony with reality on ground. African continent seems to be least affected by COVID-19. Therefore, the authors need to decide whether this study should focus on the black populations globally or be narrowed to those in western countries.
Response: Thank you for this feedback. As you have rightly pointed out, The continent of Africa was not as impacted by COVID-19 compared to other continents. However, some Black populations living outside of Africa were impacted. As highlighted in the manuscript, we systematically reviewed the intervention strategies that have been used to address COVID-19 vaccine hesitancy among Black populations across the globe and the effectiveness of these strategies or interventions. The review is not focused on studies conducted on Black populations in Africa specifically, but across continents worldwide. Based on your important suggestion, we have addressed the limited statistical reference in data from Africa. Please see lines 41,42, 61, 62, 63
Representativeness of the Data
The issue of representativeness of the data is also fundamental. The study used 14 articles and only four were carried out in the core Black countries. This raises some concerns if the results can be generalized to the Black populations globally. It would have made more meaning if the authors had just used those 10 studies from the USA and revise the topic to focus on Black populations in the USA. It is also important to note that the data were collected from different population groups with only few conducted among the large populations. One then wonders if the larger Black populations behave in the same way as healthcare workers or scientists.
Response: Thank you for the feedback. We understand that the representativeness of our research data is important and detailed attention was paid to this. The 14 studies were included because all met the study inclusion criteria - (a) addressed COVID-19 vaccine hesitancy and uptake among Black populations; (b) described an intervention or strategy for reducing COVID-19 vaccine hesitancy or described an intervention for improving COVID-19 vaccine confidence and uptake in Black communities. With limited attention to other factors such as the location or being a scientist or healthcare worker, the systematic review focused on Black community members, that is the representatives of the Black population.
Results
In line 255 and 256, the authors reported significant increase in participants’ COVID-19 vaccine intention and uptake. Was there any statistical test carried out? How do we gauge the increase as being significant without a statistical test? Will it not also be appropriate if the results are classified according to the nature of the respondents?
Response: Thank you for your observation. We have reworded the impact to 'substantial' and 'limited' based on the evident numbers and results from the included studies. We appreciate your suggestion. The results were classified according to related strategies used in the included studies to answer the research question of the systematic review - “What intervention strategies have been used to address COVID-19 vaccine hesitancy among Black populations across the globe and what is the effectiveness of these strategies or interventions?”
Conclusion
Present the limitations of the study.
Response - Thank you for the suggestion. We have addressed the limitations of the study in the discussion section. Please see lines 285-392
Reviewer 3 Report
In their manuscript entitled "Improving COVID-19 Vaccine Uptake Among Black Populations: A Systematic Review of Strategies" Adeagbo and colleagues tackle an important issue for public health. While vaccine hesitancy is not a new issues, the COVID-19 pandemic has revealed that a large percentage of the public show reluctancy toward vaccination, and different communities have different reasons for their hesitancy. Like the authors describe, the COVID-19 pandemic has disproportionately affected Black communities, largely due to inequalities, prejudice and discrimination in healthcare. These inequalities have also lead to significant mistrust of the healthcare system in black communities, which unfortunately have contributed to COVID-19 vaccine hesitancy. The current COVID-19 vaccines in the market have proven to be be greatly effective at preventing death and severe disease and thus efforts to improve COVID-19 vaccine uptake among Black populations will have a significant impact on public health and save countless lives. It is therefore, very clear, that the authors are tackling an important topic to public health. In this systematic review, the authors point out that tailored interventions that integrate culture-affirming strategies are more effective at decreasing COVID-19 vaccine hesitancy, and increasing vaccine uptake among Black populations than alternatives.
This is certainly a study of high importance to public health, and the authors have done an overall great job at developing this manuscript, however there are a few areas that need improvement. As it stands, the study reads more like a literature review, rather than a systematic review, and a bit more work is needed before to improve the overall rigor and quality of the manuscript and make it publication ready.
Comments:
1. The authors have done an amazing job at examining relevant studies regarding vaccine hesitancy in black communities, and provide here a decent review of the findings, however the critical analysis of the works seems a bit superficial. The Newcastle-Ottawa Quality is a good start for assessing study quality, but the authors should delve into the methods and interpretations of the included studies to provide a deeper critical analysis of the studies, as is pertinent for systematic reviews. This is specifically important here, as the assessment being made is qualitative.
2. The summary table is okay, but some studies are summarized with great detail, while others are somewhat vague. What are the reasons for this?
3. Going along with comment 1, the discussion section should be expanded to include a more detailed analysis of the selected studies. Mainly, what are the pros and cons of the study's approach/methods, what were their conclusions and do you agree or disagree with them based on their data. If you disagree, what is your alternative explanation/interpretation of their data? What are some potential solutions/recommendations you can make to public health officials based on the findings of this systematic review? Again, this is not simply a literature review; providing data driven recommendations and professional assessment is encouraged, and would go a long way toward improving the quality of the manuscript.
Author Response
Thank you for your feedback on our manuscript. We appreciate your comments, time and effort in improving the manuscript for publication. Please see below our point-by-point response to each of your comments:
In their manuscript entitled "Improving COVID-19 Vaccine Uptake Among Black Populations: A Systematic Review of Strategies" Adeagbo and colleagues tackle an important issue for public health. While vaccine hesitancy is not a new issues, the COVID-19 pandemic has revealed that a large percentage of the public show reluctancy toward vaccination, and different communities have different reasons for their hesitancy. Like the authors describe, the COVID-19 pandemic has disproportionately affected Black communities, largely due to inequalities, prejudice and discrimination in healthcare. These inequalities have also lead to significant mistrust of the healthcare system in black communities, which unfortunately have contributed to COVID-19 vaccine hesitancy. The current COVID-19 vaccines in the market have proven to be be greatly effective at preventing death and severe disease and thus efforts to improve COVID-19 vaccine uptake among Black populations will have a significant impact on public health and save countless lives. It is therefore, very clear, that the authors are tackling an important topic to public health. In this systematic review, the authors point out that tailored interventions that integrate culture-affirming strategies are more effective at decreasing COVID-19 vaccine hesitancy, and increasing vaccine uptake among Black populations than alternatives.
This is certainly a study of high importance to public health, and the authors have done an overall great job at developing this manuscript, however there are a few areas that need improvement. As it stands, the study reads more like a literature review, rather than a systematic review, and a bit more work is needed before to improve the overall rigor and quality of the manuscript and make it publication ready.
Comments:
1. The authors have done an amazing job at examining relevant studies regarding vaccine hesitancy in black communities, and provide here a decent review of the findings, however the critical analysis of the works seems a bit superficial. The Newcastle-Ottawa Quality is a good start for assessing study quality, but the authors should delve into the methods and interpretations of the included studies to provide a deeper critical analysis of the studies, as is pertinent for systematic reviews. This is specifically important here, as the assessment being made is qualitative.
Response: We appreciate your kind words. An overall deeper critical analysis of the review has been done, based on your suggestion. Please see below (response to comment 3)
2. The summary table is okay, but some studies are summarized with great detail, while others are somewhat vague. What are the reasons for this?
Response: Thank you for the observation. To prevent copying or reproducing the data from the included articles, we only added important facts based on the available data/information in each of the included studies.
3. Going along with comment 1, the discussion section should be expanded to include a more detailed analysis of the selected studies. Mainly, what are the pros and cons of the study's approach/methods, what were their conclusions and do you agree or disagree with them based on their data. If you disagree, what is your alternative explanation/interpretation of their data? What are some potential solutions/recommendations you can make to public health officials based on the findings of this systematic review? Again, this is not simply a literature review; providing data driven recommendations and professional assessment is encouraged, and would go a long way toward improving the quality of the manuscript.
Response: Thank you for these important questions and suggestions. We have extensively addressed all by incorporating a more detailed analysis in relevant sections in the manuscript. Please see below:
Mainly, what are the pros and cons of the study's approach/methods.
Though the studies were overwhelming quantitative, a strength of these studies was the moderate variety in the designs employed by the researchers, which collaboratively provide a broader perspective to this topic. The use of experimental approaches in [28], [36], [39], [44], [45] allowed for measurement of the effects of a specific variables on intent to vaccinate; the use of surveys in [28], [36], [39], [45], [46] facilitated data collection from a large sample, ranging from 188 to 1,353 participants among our included studies; and by employing a community-based participatory approach, Andrasik and colleagues [37] were able to meaningfully involve the community in the intervention, which directly supported their objectives of increased community engagement. There were areas of weakness also noted across these studies. Our review revealed that there is a scarcity of research on this topic, particularly outside of the United States. Additionally, the heavily quantitative nature of the studies also highlights the need for greater understanding of the perspectives and lived experiences of Black communities regarding how and why certain interventions may be more or less effective. While most studies provided a detailed account of the interventions and evaluation processes, a few provided little information as to how their strategies were employed [38], [39], [41], [42], [47]. Finally, half of our included studies did not clearly identify a specific approach, posing a challenge in assessing for methodological congruence in the undertaking of the study.
What were their conclusions and do you agree or disagree with them based on their data. If you disagree, what is your alternative explanation/interpretation of their data?
Overall, the conclusions made in our included studies were appropriate based on the designs and available data. Studies employing survey, questionnaire, or experimental approaches such as [28], [36], [39], [44], [45], [46] which included controls and comparisons allowed the researchers to directly associate their variable of interest with the various outcomes. We see from these studies that incentives are effective but have their limits; mandates increase rates of intent to vaccinate; and communication and information-based approaches are more effective when they are culturally inclusive. These findings are reflected in other non-experimental studies as well, where targeted interventions were associated with increasing enrollment or vaccination counts [35], [37], [38], [41], [43]. Two studies where the conclusions may not be transferrable were the case of Hirshberg and colleagues’ simulation [40] and Yemer and colleagues’ [47] information campaign. A simulation based on an ideal scenario may not account for naturally occurring variances in human behavior. Additionally, while Yemer and colleagues [47] concluded that the use of various communication strategies contributed to greater vaccine advocacy and increased vaccine acceptance, they provide no true evaluation methods to support this finding. In studies with no control, it is also possible that other factors contributed to the increasing counts in vaccination intent or completion, such as simply having time to consider the vaccination or weigh options. Further study on this topic is thus required to support the development of standardized interventions.
What are some potential solutions/recommendations you can make to public health officials based on the findings of this systematic review?
Though the intent to vaccinate has increased in Black American populations over time, the ongoing disparities in Western countries suggest the need for more strategic interventions to increase vaccine uptake in Black populations [31]. Our findings demonstrate the need for collaborative approaches that engage communities in identifying their priorities. Considering the importance of trust in the intent to vaccinate, community leaders should be involved at every stage of COVID-19 vaccination programs, from conceptualization to evaluation. Furthermore, in the process of developing interventions, public health officials need to consider issues related to confidence and access by asking guiding questions such as: Do people in the community want to be vaccinated? Can people in the community easily get vaccinated? How informed are people on vaccine related issues to make informed decisions? Asking these questions supports decision-making about the type of interventions that are needed in a community [50]. These questions also bring attention to the various sociocultural factors that contribute to the rates of COVID-19 vaccination in Black communities.
Round 2
Reviewer 2 Report
The authors have responded to my comments. Now I do not have any objection to its being published.